

# Seasonal variability and vertical distribution of autotrophic and heterotrophic picoplankton in the Central Red Sea

Najwa Al-Otaibi[1], Tamara M. Huete-Stauffer[1], Maria Ll. Calleja[2], Xabier Irigoien[3,4] and Xosé Anxelu G. Morán[1]

[1] Red Sea Research Center (RSRC), Division of Biological and Environmental Sciences and Engineering, King Abdullah University of Science and Technology (KAUST), Thuwal, Saudi Arabia
[2] Department of Climate Geochemistry, Max Planck Institute for Chemistry (MPIC), Mainz, Germany
[3] AZTI - Marine Research, Pasaia, Spain
[4] Basque Foundation for Science, IKERBASQUE, Bilbao, Spain

Corresponding author
Xosé Anxelu G. Morán,
xelu.moran@kaust.edu.sa

## ABSTRACT

The Red Sea is characterized by higher temperatures and salinities than other oligotrophic tropical regions. Here, we investigated the vertical and seasonal variations in the abundance and biomass of autotrophic and heterotrophic picoplankton. Using flow cytometry, we consistently observed five groups of autotrophs (*Prochlorococcus*, two populations of *Synechococcus* separated by their relative phycoerythrin fluorescence, low (LF-Syn) and high (HF-Syn), and two differently-sized groups of picoeukaryotes, small (Speuk) and large (Lpeuk)) and two groups of heterotrophic prokaryotes of low and high nucleic acid content (LNA and HNA, respectively). Samples were collected in 15 surveys conducted from 2015 to 2017 at a 700-m depth station in the central Red Sea. Surface temperature ranged from 24.6 to 32.6 °C with a constant value of 21.7 °C below 200 m. Integrated (0–100 m) chlorophyll *a* concentrations were low, with maximum values in fall (24.0 ± 2.7 mg m$^{-2}$) and minima in spring and summer (16.1 ± 1.9 and 1.1 mg m$^{-2}$, respectively). Picoplankton abundance was generally lower than in other tropical environments. Vertical distributions differed for each group, with *Synechococcus* and LNA prokaryotes more abundant at the surface while *Prochlorococcus*, picoeukaryotes and HNA prokaryotes peaked at the deep chlorophyll maximum, located between 40 and 76 m. Surface to 100 m depth-weighted abundances exhibited clear seasonal patterns for *Prochlorococcus,* with maxima in summer (7.83 × 10$^4$ cells mL$^{-1}$, July 2015) and minima in winter (1.39 × 10$^4$ cells mL$^{-1}$, January 2015). LF-Syn (0.32 – 2.70 × 10$^4$ cells mL$^{-1}$), HF-Syn (1.11 – 3.20 × 10$^4$ cells mL$^{-1}$) and Speuk (0.99 – 4.81 × 10$^2$ cells mL$^{-1}$) showed an inverse pattern to *Prochlorococcus,* while Lpeuk (0.16 – 7.05 × 10$^4$ cells mL$^{-1}$) peaked in fall. *Synechococcus* unexpectedly outnumbered *Prochlorococcus* in winter and at the end of fall. The seasonality of heterotrophic prokaryotes (2.29 – 4.21×10$^5$ cells mL$^{-1}$) was less noticeable than autotrophic picoplankton. The contribution of HNA cells was generally low in the upper layers, ranging from 36% in late spring and early summer to ca. 50% in winter and fall. Autotrophs dominated integrated picoplankton biomass in the upper 100 m, with 1.4-fold higher values in summer than in winter (mean 387 and 272 mg C m$^{-2}$, respectively). However, when the whole water column was considered,

the biomass of heterotrophic prokaryotes exceeded that of autotrophic picoplankton with an average of 411 mg C m$^{-2}$. Despite being located in tropical waters, our results show that the picoplankton community seasonal differences in the central Red Sea are not fundamentally different from higher latitude regions.

## INTRODUCTION

Picoplankton comprises both autotrophic and heterotrophic unicellular organisms in the size range of 0.2 to 2 μm. Picocyanobacteria of the genera *Prochlorococcus* (typically 0.6–0.8 μm in diameter) and *Synechococcus* (ca. 1 μm) usually dominate numerically autotrophic picoplankton, which also includes a high diversity of picoeukaryotes larger than 1 μm (*Campbell et al., 1997*; *Giovannoni & Vergin, 2012*). *Prochlorococcus* is usually more abundant than *Synechococcus* in highly stratified and low-nutrient surface waters (*DuRand, Olson & Chisholm, 2001*; *Giovannoni & Vergin, 2012*; *Olson et al., 1990*; *Zubkov et al., 2000*; *Zwirglmaier et al., 2007*). Picoeukaryotes are less abundant than picocyanobacteria (*Kirkham et al., 2013*), especially in the tropical and subtropical oceans (*Kirkham et al., 2013*; *Morán, Fernández & Pérez, 2004*). Heterotrophic picoplankton are mostly prokaryotes, overwhelmingly dominated by bacteria over archaea in the upper layers since the abundance of the latter only increases significantly at depth (*Karner, DeLong & Karl, 2001*). Two flow cytometric populations of heterotrophic prokaryotes are typically detected after staining with DNA-binding dyes: high (HNA) and low (LNA) nucleic acid content cells (*Gasol et al., 1999*; *Li, Jellett & Dickie, 1995*; *Nishimura, Kim & Nagata, 2005*; *Sherr, Sherr & Longnecker, 2006*). The HNA group typically dominates in eutrophic and mesotrophic conditions characterizing the colder, nutrient-rich months while LNA tends to dominate in stratified oligotrophic environments (*Calvo-Díaz & Morán, 2006*; *Morán et al., 2007*; *Zubkov, Allen & Fuchs, 2004*).

Seasonal changes in the abundance of autotrophic picoplankton groups are well known in temperate (*Calvo-Díaz & Morán, 2006*; *Li, 1998*; *Morán, 2007*) and polar waters (*Iversen & Seuthe, 2011*; *Rivkin, 1991*) while they are less known in lower latitude waters, with the exception of two long-term sites: the Bermuda Atlantic Time Series (BATS) in the western Sargasso Sea (*DuRand, Olson & Chisholm, 2001*; *Malmstrom et al., 2010*) and the Hawaii Ocean Time-series (HOT) in the North Pacific subtropical gyre (*Campbell et al., 1997*; *Malmstrom et al., 2010*). In contrast to autotrophs, the seasonality of heterotrophic bacteria in subtropical and tropical waters is thought to be less pronounced than in temperate regions (*Bunse & Pinhassi, 2017*).

The Red Sea is an oligotrophic marine basin with very high temperatures (up to 35 ° C at the surface in summer, *Calbet et al., 2015*; *Chaidez et al., 2017*; *Rasul, Stewart & Nawab, 2015*) and salinities (ca. 40, *Tesfamichael & Pauly, 2016*). The effect of these quasi-extreme conditions on the seasonality of picoplankton communities has received

far less attention than other oligotrophic waters. Understanding the temporal changes of picoplankton abundance and their response to environmental drivers are essential to define the lower trophic levels of Red Sea pelagic food webs. Regarding autotrophs, we have a good understanding of their seasonal variability (*Al-Najjar et al., 2007*; *Lindell & Post, 1995*; *Post et al., 2011*) and their trophic relationships with other components of the microbial food web in the northern reaches, especially in the Gulf of Aqaba (*Berninger & Wickham, 2005*; *Sommer, 2000*; *Sommer et al., 2002*). For heterotrophic prokaryotes, although our knowledge about their diversity is increasing (*Ngugi et al., 2012*; *Pearman et al., 2017*; *Thompson et al., 2017*), only a few studies have investigated their vertical distribution in Red Sea waters (*Calbet et al., 2015*; *Qian et al., 2011*). A recent report using data collected from the same site as this study has shown that the abundance of heterotrophic bacteria can change temporally up to 3-fold within the same depth in the upper epipelagic (*Calleja, Al-Otaibi & Morán, 2019*; *García et al., 2018*). Other studies conducted at that site have shown that LNA bacteria dominated in the epipelagic layer, while HNA cells were more abundant in the mesopelagic layer, indicating that each group seems to prefer different environmental conditions (*Calleja, Al-Otaibi & Morán, 2019*; *Calleja et al., 2018*). The unexpectedly low standing stocks of heterotrophic bacteria in a nearby shallow embayment have been explained by strong top-down control exerted by protistan grazers and viruses (Sabbagh et al. submitted; *Silva et al., 2019*).

Here, we conducted a detailed investigation of both the temporal and vertical variability of autotrophic and heterotrophic picoplankton, as assessed by flow cytometry, by periodic sampling over two years (2015–2017) at a mesopelagic station (ca. 700 m depth) in the central Red Sea, Saudi Arabia. Given the tropical characteristics of the site, we hypothesize that the seasonal variability of picoplankton in epipelagic waters would be lower than that found at higher latitudes, and the marked stratification found between 100 and 200 m should result in strong vertical gradients in abundance, size and ultimately biomass.

## MATERIALS & METHODS

### Sample collection and environmental properties

Periodic samplings were conducted from January 2015 to May 2017 on board of RV Thuwal at a mesopelagic station (ca. 700 m depth) located in the central Red Sea, 6 km off the coast of King Abdullah Economic City (KAEC) in Saudi Arabia. Sailing permission were approved by Saudi Coast Guard. We performed 15 vertical profiles evenly distributed along the four seasons (only winter had 3 samples rather than 4, Table 1). Samples were taken at regular depths from the surface to the bottom: 5, 20, 40–80 targeting the deep chlorophyll maximum (DCM), 100, 200, 300, 400, 550, 600 and 700 m. Temperature, salinity, fluorescence and photosynthetically active radiation (PAR) data were acquired with SeaBird SB9 Plus or IDRONAUT 305 CTDs. PAR was available for only 7 sampling times. The depth of the photic layer was determined by the vertical light attenuation coefficient ($Kd$) as the depth receiving 1% of surface irradiance (*Calvo-Díaz & Morán, 2006*). Stratification index (SI) was calculated as the density at 100 m minus that at the surface (*Calvo-Díaz & Morán, 2006*). The upper mixed layer (UML) depth was determined

**Table 1  Seasonal distribution and date of the 15 individual samplings at the study site, with the corresponding day of the year for the assessment of seasonal patterns.**

| Season | Sampling date (dd/mm/yyyy) | Day of year |
|---|---|---|
| Winter | 19/01/2015 | 18 |
|  | 02/02/2016 | 32 |
|  | 25/02/2017 | 55 |
| Spring | 24/03/2015 | 82 |
|  | 06/03/2016 | 65 |
|  | 24/04/2017 | 113 |
|  | 5/22/2017 | 141 |
| Summer | 01/07/2015 | 181 |
|  | 25/08/2015 | 236 |
|  | 05/09/2015 | 247 |
|  | 21/06/2016 | 172 |
| Fall | 26/10/2015 | 298 |
|  | 11/11/2015 | 314 |
|  | 09/12/2015 | 342 |
|  | 10/26/2016 | 299 |

as the first depth in which the difference in density with the shallower 5 m was $\geq 0.05$ kg m$^{-3}$ (*Calvo-Díaz & Morán, 2006*).

Water samples were taken from Niskin bottles in a rosette sampler with an attached CTD probe (Fig. S1). In 2015 total chlorophyll *a* concentration (Chl *a*) was obtained after filtering 500 to 2,000 ml of the sample through Whatman GF/F filters (25 mm diameter). After checking for the minimum volume yielding reliable results, in 2016 and 2017 we performed sequential filtration of 200 ml samples through filters of 20, 2 and 0.2 µm of pore-size (IsoporeTM Membrane Filters, RTTP, 47 mm diameter), so that Chl *a* was the sum of the corresponding size-fractions: micro- (above 20 µm), nano- (between 2 and 20) and picophytoplankton (between 0.2 and 2 µm). Filters were frozen at −80 °C until analysis in the laboratory. Pigments were extracted in 90% acetone for 24 h in the dark at 4 °C and chlorophyll *a* fluorescence was measured with a Trilogy fluorometer (Turner) calibrated with pure extracts.

Samples for dissolved inorganic nitrogen (DIN = $NO^{3-}$ + $NO^{2-}$), dissolved inorganic phosphorus (DIP = $PO_4^{3-}$), dissolved organic carbon (DOC) and total dissolved nitrogen (TDN) were filtered through pre-combusted GF/F filters and analyzed as previously reported by *Calleja et al. (2018)*. The nutricline depth was defined as the depth where nitrate concentration first reached 1 µmol L$^{-1}$ (*Calvo-Díaz & Morán, 2006*).

## Analysis of picoplankton by flow cytometry

Picoplankton samples (1.8 mL) were preserved with 1% paraformaldehyde + 0.05% glutaraldehyde final concentration and placed in the dark for approximately 10 min, then frozen in liquid nitrogen and stored at −80 °C once in the laboratory. After thawing, samples were analyzed with a FACSCanto II flow cytometer (BD-Biosciences). Molecular

Probes fluorescent latex beads of 1 μm were used as an internal standard for size and fluorescence measurements. We analyzed aliquots of 0.6 mL for autotrophs and 0.4 mL for heterotrophs, at high (mean 117.9 μL min$^{-1}$) and low (17.9 μL min$^{-1}$) flow rates, respectively, until acquiring 10,000 events. Before analysis, heterotrophic bacteria were stained with 2.5 μmol L$^{-1}$ of the DNA fluorochrome SYBR Green II (*Gasol & Morán, 2015*). All cytograms were analyzed with FCSExpress 5 software. Autotrophic prokaryotic cells were classified as cyanobacteria (*Synechococcus* and *Prochlorococcus*) and picoeukaryotes according to their orange (PE, 433 nm) and red (PerCP-Cy5-5, 498 nm) fluorescence and light scatter at 90° or side scatter (SSC) signals. Two groups of heterotrophic prokaryotes were distinguished based on their relative green fluorescence (FITC, 360 nm) signal: low and high nucleic acid content (LNA and HNA, respectively). Cell size was determined by an empirical calibration between relative SSC and cell diameter according to *Calvo-Díaz & Morán (2006)*. Spherical shape was assumed for all groups for estimating biovolume, which was transformed into individual biomass by using the biovolume-to-carbon conversion factor of 237 fg C μm$^{-3}$ for autotrophs (*Worden, Nolan & Palenik, 2004*) and the equation biomass = 108. 8× (biovolume)$^{0.898}$ for heterotrophs (*Gundersen et al., 2002*). The biomass of each picoplanktonic group was finally obtained by multiplying the individual biomass estimate by the corresponding abundance.

## Statistical analyses

Picoplankton abundance, biovolume and biomass data were log10-transformed to attain normality and assess their relationship with environmental variables by Spearman's rank correlation coefficient. One-way ANOVAs and post hoc Tukey's pairwise comparisons were used to determine significant variations between seasons ($P < 0.05$) with OriginPro software. A non-metric multidimensional scaling (NMDS), a distance-based ordination technique, was performed on the Bray-Curtis dissimilarity distances together with pairwise PERMANOVAs in order to summarize the seasonal and vertical changes in the abundance of the different picoplankton groups and their relation with environmental variables in the upper epipelagic zone. Four groups of samples were considered according to depth: surface, above DCM, DCM depth, below DCM and 100 m. NMDS stress values, a measure of goodness-of-fit, can be used to evaluate the proper choice of dimensions. Low values (0.05–0.1) provide a good fit in reduced dimensions while values >0.3 indicate that the ordination is arbitrary and potentially uninterpretable (*Ramette, 2007*; *Zhu & Yu, 2009*). The NMDS analysis was done in R (http://www.r-project.org) and we used the "envfit" function in order to estimate the correlations between the environmental variables and the NMDS axis scores.

## RESULTS

### Vertical and seasonal variability in hydrographic conditions

The mean vertical profiles of selected environmental variables for the four seasons are shown in Fig. 1. As expected, significant differences (ANOVA: $F = 14.4$, $p = 0.0004$, $n = 15$) in mean surface temperature were found, with summer values 6.3 °C higher than in winter (Table 2). The temperature remained constant year-round from 200 m down

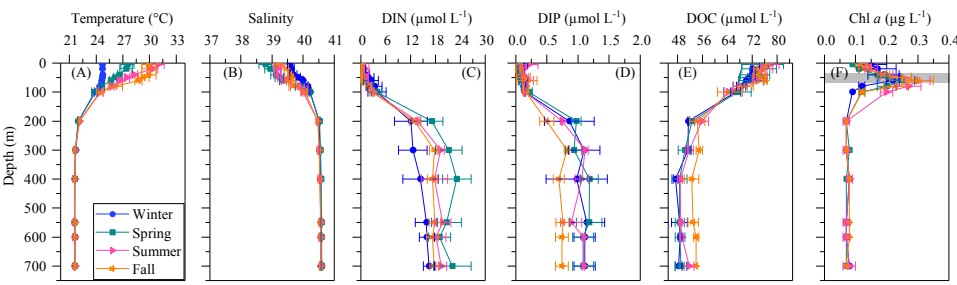

**Figure 1 Mean seasonal vertical profiles of environmental variables.** (A) temperature, (B) salinity, (C) dissolved inorganic nitrogen (nitrate + nitrite, DIN), (D) dissolved inorganic phosphorus (DIP), (E) dissolved organic carbon (DOC), and (F) chlorophyll *a* (Chl *a*) concentrations in winter, spring, summer and fall at the study station. Error bars show the standard error of the mean. The shaded area in (F) indicates the overall range of the deep chlorophyll maximum (DCM). See the text for details.

to the bottom at 21.7 ± 0.02 °C SE (Fig. 1A). Surface salinity displayed slight seasonal variations from 38.8 ± 0.2 in spring to 39.6 ± 0.1 in winter (Table 2), but there was no seasonal difference below 200 m (40.6 ± 0.0). Differences in SI were not significant despite some seasonality (Table 2), but the UML was significantly shallower in summer than in the other seasons (Table 2, ANOVA: $F = 18.3$, $p = 0.0002$, $n = 14$). The euphotic layer depth varied from 63 to 89 m, with similar mean values across seasons (Table 2).

Dissolved inorganic nitrogen (DIN = nitrate + nitrite) presented uniformly low concentrations at the surface (0.17 ± 0.11 µmol $L^{-1}$) but reached 20.5 ± 1.6 µmol $L^{-1}$ at depths higher than 200 m (Fig. 1C), with an average nutricline depth of 67 ± 6 m (Table 2). Dissolved inorganic phosphate (DIP) followed the same pattern as DIN, with low seasonal mean values at the surface (0.10 ± 0.04 µmol $L^{-1}$) and increasing with depth to a seasonal mean maximum of 1.17 ± 0.1 µmol $L^{-1}$at around 600 m depth (Fig. 1D). The concentration of dissolved organic carbon (DOC, Fig. 1E) declined with depth from a mean 76.1 ± 7.5 µmol $L^{-1}$ at the surface to 52.2 ± 5.8 µmol $L^{-1}$ below 200 m.

## Total and size-fractionated chlorophyll *a* concentration

The vertical distribution of total chlorophyll *a* (Chl *a*) concentration (Fig. 1F) showed a consistent and clear deep chlorophyll maximum (DCM) located at an average depth of 56 ± 4 m (Table 2). Surface seasonal mean values ranged from 0.09 to 0.15 µg $L^{-1}$ (Fig. 1F). Mean integrated Chl *a* values for the upper 100 m increased from 16.1 mg $m^{-2}$in spring and summer (±2.06 and 1.09, respectively) to 19.0 ± 2.9 mg $m^{-2}$ in winter and 24.0 ± 2.7 mg $m^{-2}$ in fall (Fig. 2A). The picoplankton size fraction contributed, on average, 70.8 ± 1.0% to total integrated values, with nanoplankton and microplankton making up 21.9 ± 1.5% and 7.3 ± 1.9%, respectively, with no significant differences in the relative contributions of the three size classes (Fig. 2B).

## Vertical distribution of picoplankton abundance and cellular characteristics

*Prochlorococcus*, *Synechococcus* and picoeukaryotes were mostly restricted to the upper 100 m, with none of the groups detected in significant numbers at or below 150 m

**Table 2** **Average seasonal values of environmental properties at the surface of the study site and characteristic depths (mean ± SE).** Sea surface temperature (SST), salinity, dissolved inorganic nitrogen (nitrate + nitrite, DIN), dissolved inorganic phosphorus (phosphate, DIP), dissolved organic carbon (DOC), total chlorophyll *a* concentration (Chl *a*), stratification index (SI) and depths of the upper mixed layer (UML), the euphotic zone (Zph), the deep chlorophyll maximum (DCM) and the nutricline (NC). Stars and superscript letters indicate significant differences between seasons (ANOVA and Tukey post hoc test; *, $p = 0.05$; ** $p = 0.01$; *** $p = 0.001$.

| | Winter | Spring | Summer | Fall |
|---|---|---|---|---|
| **SST *** (°C)** | $24.7 \pm 0.1$ [a] | $27.5 \pm 0.7$ [a,c] | $30.9 \pm 0.7$ [b,c] | $29.9 \pm 0.8$ [c] |
| | $n = 3$ | $n = 4$ | $n = 4$ | $n = 4$ |
| **Salinity** | $39.6 \pm 0.1$ | $38.8 \pm 0.2$ | $39.2 \pm 0.2$ | $39.2 \pm 0.2$ |
| | $n = 3$ | $n = 4$ | $n = 4$ | $n = 4$ |
| **DIN ($\mu$mol l$^{-1}$)** | $0.1 \pm 0.04$ | $0.2 \pm 0.1$ | $0.3 \pm 0.2$ | $0.2 \pm 0.1$ |
| | $n = 2$ | $n = 4$ | $n = 2$ | $n = 3$ |
| **DIP ($\mu$mol l$^{-1}$)** | $0.1 \pm 0.1$ | $0.04 \pm 0.01$ | $0.2 \pm 0.1$ | $0.1 \pm 0.04$ |
| | $n = 2$ | $n = 4$ | $n = 2$ | $n = 3$ |
| **DOC ($\mu$mol l$^{-1}$)** | $75.8 \pm 1.4$ | $75.4 \pm 6.7$ | $77.7 \pm 2.9$ | $75.3 \pm 1.7$ |
| | $n = 2$ | $n = 4$ | $n = 4$ | $n = 3$ |
| **Chl *a* ($\mu$g l$^{-1}$)** | $0.15 \pm 0.05$ | $0.09 \pm 0.003$ | $0.13 \pm 0.02$ | $0.12 \pm 0.02$ |
| | $n = 2$ | $n = 4$ | $n = 4$ | $n = 4$ |
| **SI** | $1.03 \pm 0.14$ | $2.68 \pm 0.52$ | $2.74 \pm 0.27$ | $2.39 \pm 0.26$ |
| | $n = 2$ | $n = 4$ | $n = 4$ | $n = 4$ |
| **UML *** (m)** | $60 \pm 7$ [a] | $43 \pm 2$ [a] | $19 \pm 3$ [b] | $48 \pm 5$ [a] |
| | $n = 2$ | $n = 4$ | $n = 4$ | $n = 4$ |
| **Zph (m)** | $85 \pm 1$ | $85 \pm 2$ | $76 \pm 4$ | $72 \pm 4$ |
| | $n = 2$ | $n = 3$ | $n = 4$ | $n = 4$ |
| **DCM (m)** | $55 \pm 1$ | $62 \pm 8$ | $63 \pm 8$ | $44 \pm 12$ |
| | $n = 4$ | $n = 4$ | $n = 4$ | $n = 4$ |
| **NC (m)** | $63 \pm 7$ | $74 \pm 8$ | $47 \pm 35$ | $75 \pm 5$ |
| | $n = 2$ | $n = 4$ | $n = 2$ | $n = 3$ |

depth. Figure 3 shows the average vertical distribution of picophytoplankton abundance, cell size and relative red fluorescence (as a proxy of Chl *a* content) for each season. *Prochlorococcus* abundance was generally low at the surface ($1.11–5.81 \times 10^4$ cells mL$^{-1}$) and peaked at the DCM ($1.32 \pm 0.16 \times 10^5$ cells mL$^{-1}$ in summer) (Fig. 3A). The two groups of *Synechococcus* discriminated by low (LF-Syn) and high (HF-Syn) phycoerythrin fluorescence were consistently less abundant than *Prochlorococcus*. LF-Syn and HF-Syn tended to show higher numbers in the surface layers, with averages of $2.29 \pm 0.53 \times 10^4$ and $3.47 \pm 0.54 \times 10^4$ cells mL$^{-1}$, respectively (Figs. 3B and 3C). HF-Syn reached deeper than LF-Syn with the latter virtually absent at 80 m (Figs. 3B and 3C). Two groups of picoeukaryotes according to size were consistently distinguished, hereafter referred to as Small (Speuk) and large (Lpeuk). Speuk vertical distribution was similar to that of *Prochlorococcus* (Fig. 3D), while Lpeuk usually disappeared deeper than 40–60 m except in fall, where the highest values were found in the DCM (Figs. 3D and 3E). Coincident with declining abundances, the biovolume of all groups increased steadily with depth from 40 m downwards except for Lpeuk (Figs. 3F–3J). Similar to biovolume, the relative red

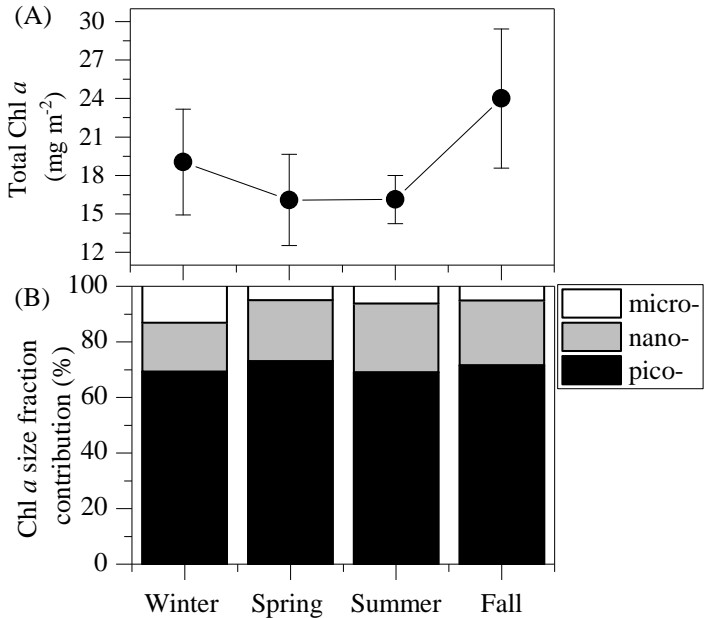

**Figure 2** **Mean seasonal values of total and size-fractionated chlorophyll *a* concentration.** (A) total integrated chlorophyll *a* concentration and (B) mean contributions of the three size-fractions in the upper 100 m of the study station. pico- : picoplankton, nano- : nanoplankton and micro- : microplankton.

fluorescence increased consistently with depth, with less marked patterns for LF-Syn and Lpeuk due to their shallower distribution (Figs. 3K–3O).

The mean seasonal distribution of heterotrophic prokaryotes abundance and cell size with depth is shown in Fig. 4. The abundances of both LNA and HNA cells were highest in the upper 100 m (maxima of $2.92 \times 10^5$ and $2.51 \times 10^5$ cells mL$^{-1}$a t 20 and 40 m, respectively) but remained relatively stable for the entire mesopelagic layer (Figs. 4A and 4B). LNA were more abundant than HNA cells in the upper epipelagic, resulting in a contribution of HNA cells to total abundance (%HNA) that ranged from 38.3 to 47.1% at the surface. Values increased to 52.3 –57.4% at 200 m and remained pretty homogeneous down to the sea floor (Fig. 4C). The biovolume of HNA cells was consistently larger than that of LNA cells throughout the water column. Differences were observed between seasons, with maxima in winter for both groups and minima in summer for HNA and in fall for LNA cells (Figs. 4D and 4E).

## Seasonal variation of picoplankton abundance and cellular characteristics

Although some differences between seasons were already apparent in the vertical distributions (Figs. 3 and 4), depth-weighted averages from the surface to 100 m were calculated to better capture the seasonal changes. *Prochlorococcus* abundance displayed a clear seasonal pattern, with minimum values in winter ($1.4 \times 10^4$ cells mL$^{-1}$, January 2015) and maximum values in summer ($7.8 \times 10^4$ cells mL$^{-1}$, July 2015) (Fig. 5A). The two groups of *Synechococcus* shared similar dynamics, with maximum values in spring

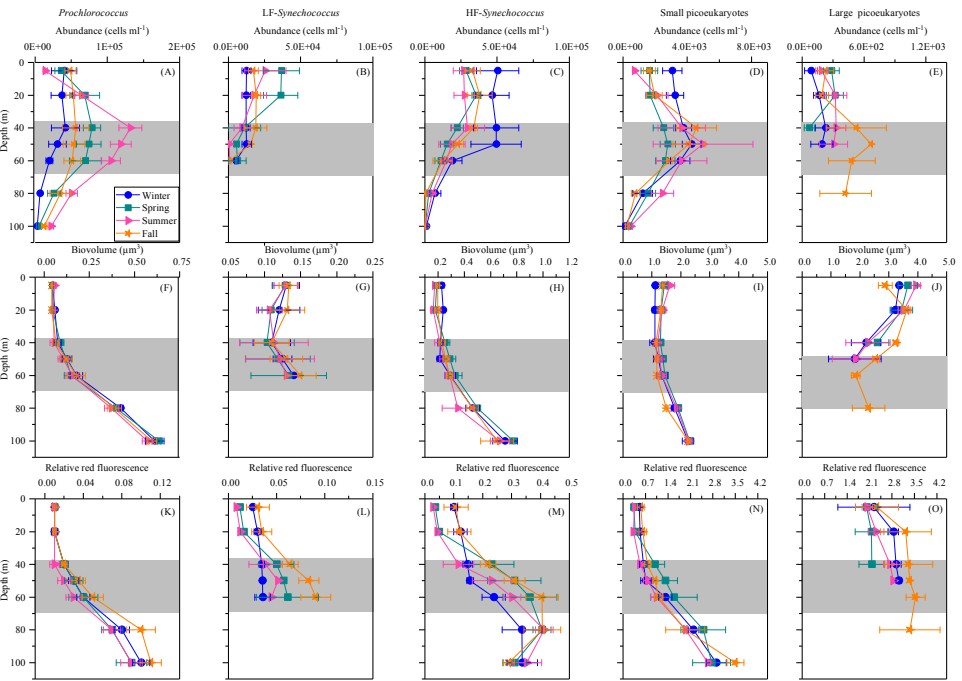

**Figure 3** **Vertical distribution of autotrophic picoplankton mean seasonal abundance and cellular characteristics.** (A–E) abundance, (F–J) biovolume,(K–O) relative red fluorescence of *Prochlorococcus*, low (LF-Syn) and high (HF-Syn) phycoerythrin fluorescence populations of *Synechococcus* and small and large picoeukaryotes in winter, spring, summer and fall at the study station. Error bars show the standard error of the mean. The shaded area indicates the overall range of the DCM.

for LF-Syn ($2.7 \times 10^4$ cells mL$^{-1}$, March 2015) and winter for HF-Syn ($3.2 \times 10^4$ cells mL$^{-1}$, January 2015) while the lowest values were observed between late spring and early summer for both groups (Fig. 5B). Consequently, the ratio between *Prochlorococcus* and *Synechococcus* abundance, which was higher than 1 for most of the year, was occasionally lower in winter and at the end of fall (Fig. 5C). Speuk abundance presented low values in summer ($9.9 \times 10^2$ cells mL$^{-1}$, July 2015) and higher in winter ($3.60 \times 10^3$ cells mL$^{-1}$, January 2015) while Lpeuk, generally less abundant, peaked in fall ($7.1 \times 10^2$ cells mL$^{-1}$, October 2016) (Fig. 5D). On an annual basis, *Prochlorococcus* contributed 57.6 ± 4.2% to total picophytoplankton cell numbers, followed by *Synechococcus* (38.9 ± 3.9%) with picoeukaryotes 21- to 79-fold lower abundances than cyanobacteria. Depth-weighted biovolumes (0.14–0.25 µm$^3$ *Prochlorococcus*, 0.08–0.19 µm$^3$ LF-Syn, 0.27–0.42 µm$^3$ HF-Syn, 1.09–1.66 µm$^3$ Speuk, 2.50–3.62 µm$^3$ Lpeuk) did not show any clear seasonal pattern with slightly increased values of Speuk in early summer and Lpeuk in fall (Figs. S2A–S2C). The seasonality of 0–100 m mean relative red fluorescence as a proxy for Chl *a* content followed the expected summer minimum only for *Prochlorococcus* and LF-Syn (Figs. S2D–S2F). Differences in biovolume affected little the changes mentioned above in abundance when calculating the biomass of the different picophytoplankton groups. Integrated autotrophic picoplankton biomass for the upper 100 m showed higher values in summer (387.4 mg C m$^{-2}$) with a significant contribution of *Prochlorococcus* (46.6 ± 6%)

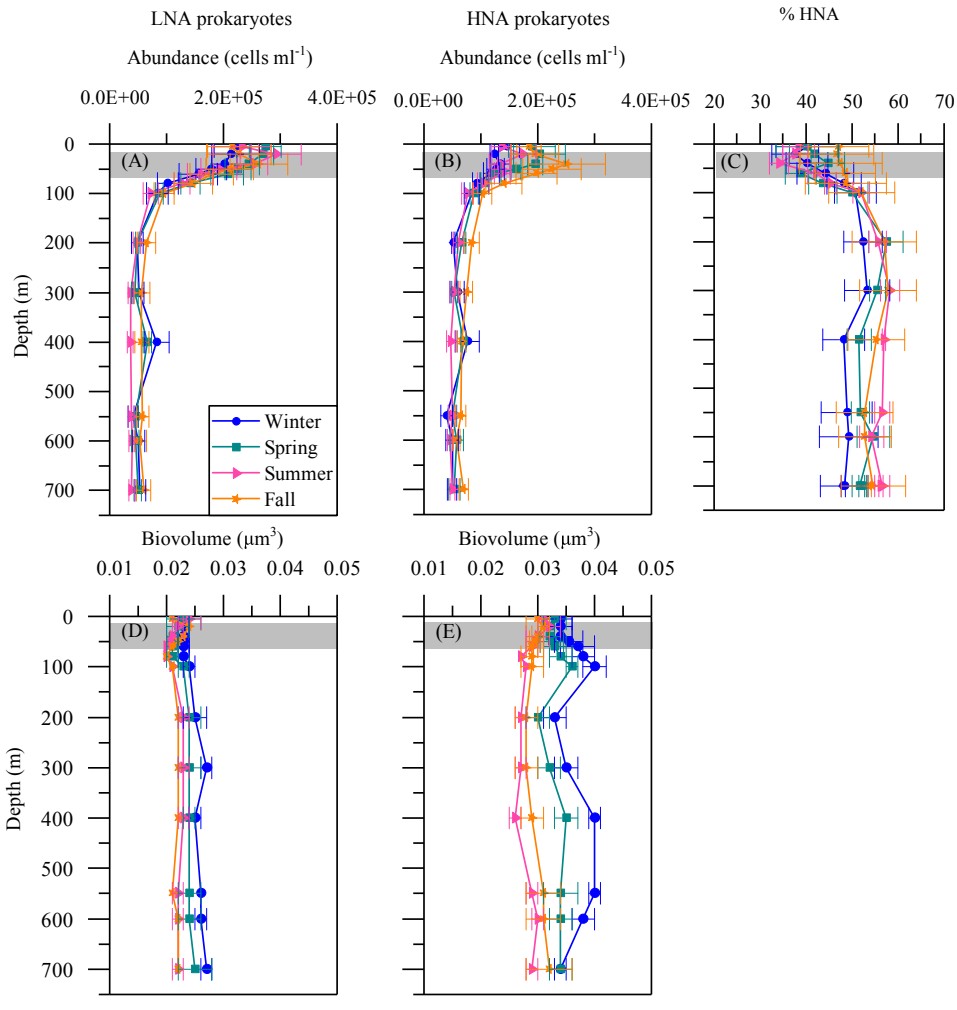

**Figure 4** **Vertical distribution of heterotrophic prokaryotes mean seasonal abundance and biovolume.** Abundance of LNA (A) and (B) HNA cells, (C) contribution of HNA cells to total numbers (%HNA) and biovolume of LNA (D) and HNA (E) cells in winter, spring, summer and fall at the study station. Error bars show the standard error of the mean. The shaded area indicates the overall range of the DCM.

(ANOVA: $F = 4.2$, $p = 0.03$, $n = 15$) except in winter when *Synechococcus* contributed 49.5% with a high contribution of HF-Syn (37.02%).

The mean total abundance of heterotrophic prokaryotes (HNA + LNA prokaryotes) in the upper 100 m ranged from 2.29 to 4.21 × $10^5$ cells mL$^{-1}$, with higher values in spring and fall and lower in summer (Dataset S1). Figure 6A shows the corresponding values for the LNA and HNA groups with respective annual means of $1.87 \pm 0.01 \times 10^5$ and $1.38 \pm 0.07 \times 10^5$ cells mL$^{-1}$. Although their abundances failed to show marked seasonal patterns, a clear seasonality in the contribution of HNA bacteria emerged. Upper epipelagic-averaged %HNA values ranged from 35.9% in late spring and early summer to ca. 50% in winter and fall (Fig. 6B). In contrast, the seasonality in biovolume and relative nucleic acid content was not clear for any of the two groups (Figs. S3A and S3B). The

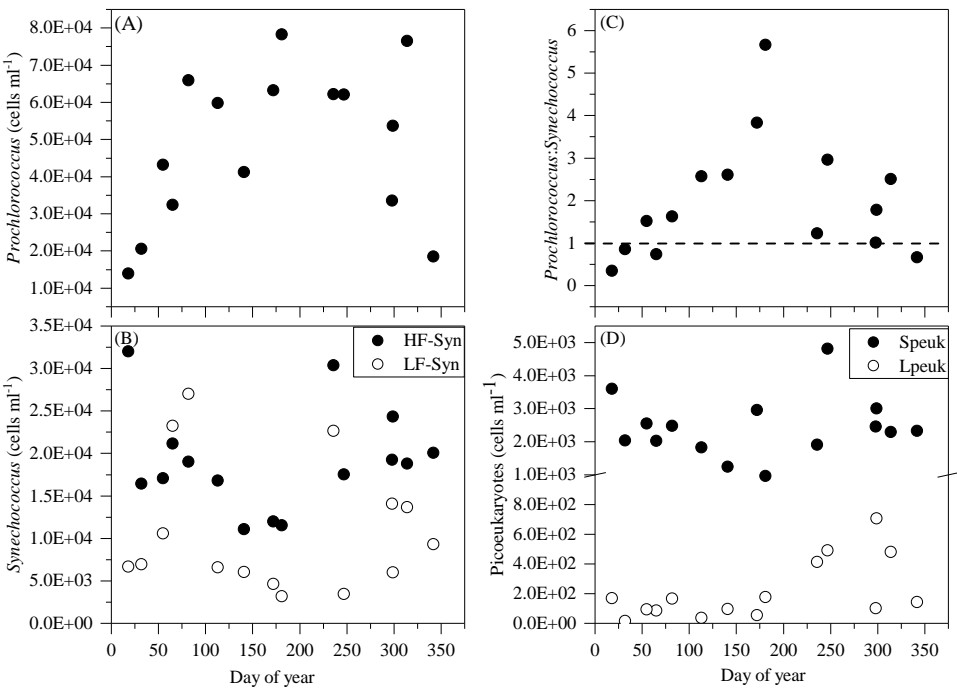

**Figure 5** Temporal variability of autotrophic picoplankton abundances averaged for the upper 100 m. (A) *Prochlorococcus*, (B) high (HF-Syn) and low fluorescence (LF-Syn) *Synechococcus* and (D) small (Speuk) and large (Lpeuk) picoeukaryotes. Also shown in (C) the ratio between *Prochlorococcus* and *Synechococcus* cell abundances.

integrated biomass of heterotrophic prokaryotes in the upper 100 m ranged from 86.9 to 257.6 mg C m$^{-2}$, with seasonal means shown in Fig. 7. Regarding the contribution of LNA and HNA cells to total heterotrophic prokaryotes biomass in the upper epipelagic, differences were minor with the HNA group prevailing in winter (53.8%) and fall (51.3%) and the LNA groups in summer (53.4%) and spring (52.3%) (Fig. 7). Overall, the biomass of autotrophic picoplankton groups was consistently higher than that of heterotrophic bacteria in the upper epipelagic, with annual averages of 348.1 ± 20.5 and 140.8 ± 10.1 mg C m$^{-2}$, respectively. However, when values were integrated over the entire water column (0–700 m), the mean biomass of heterotrophic bacteria increased to 410.7 ± 27.0 mg C m$^{-2}$, thus exceeding the total biomass of autotrophic picoplankton (Fig. 7). Moreover, when the entire water column was considered HNA cells clearly dominated total biomass regardless of the season.

## Relationships with environmental variables

Figure 8 shows the NMDS performed on the Bray-Curtis distances of the abundances for autotrophic and heterotrophic picoplankton populations at different depths in the upper epipelagic (Fig. 8). The low stress value (0.1) indicated a reliable distribution of the samples in two dimensions. All environmental variables were initially considered in the NMDS analysis, but some of them (e.g., DIP, DOC, UML, etc.) were removed since they did not show significant effects on the distribution of the samples. The correlation of
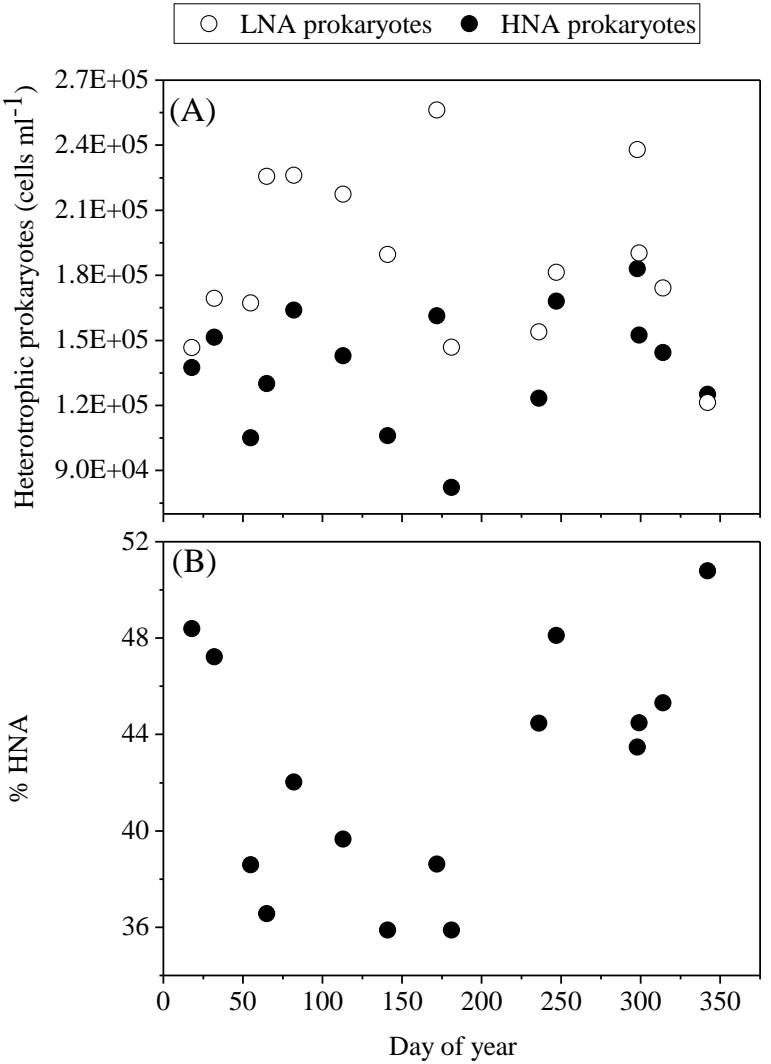

**Figure 6 Temporal variability of heterotrophic prokaryotes abundances averaged for the upper 100 m.**
(A) low (LNA) and high (HNA) nucleic acid bacteria and (B) contribution of HNA cells to total numbers
(%HNA).

the NMDS scores (position of the samples) with the environmental variables (represented
by the arrows) indicated significant effects of temperature ($r = 0.66$, $p = 0.002$), DON
($r = 0.48$, $p = 0.034$, Chl $a$ ($r = 0.62$, $p = 0.003$), DIN ($r = 0.72$, $p = 0.001$) and salinity
($r = 0.75$, $p = 0.001$). The NMDS plot also showed different habitat segregation of the
picoplanktonic groups, with an overall significant effect of the depth layer (PERMANOVA:
$r^2 = 0.53$, $p < 0.01$). The most abundant group in the surface was *Synechococcus* (mostly
the HF_Syn), where temperature and DON were highest. In contrast, *Prochlorococcus*
and picoeukaryotes (mainly Speuk) were more abundant around the depth of the DCM,
actually contributing to the increase in Chl *a*. HNA and LNA had a higher weight at 100
m, primarily because of the decrease in autotrophic picoplankton groups, where DIN and

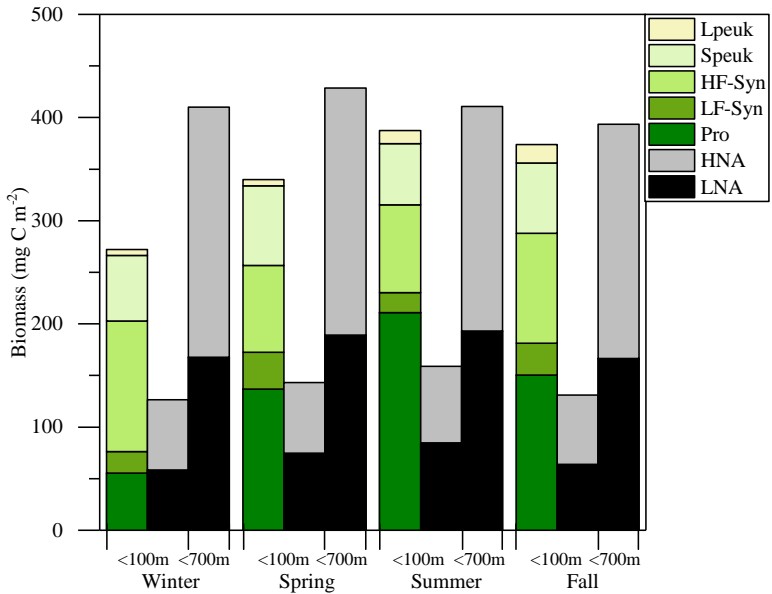

**Figure 7 Mean seasonal values of autotrophic and heterotrophic picoplankton integrated biomass in winter, spring, summer and fall at the study station.** The green-yellow bar shows the integrated biomass for the upper 100 m of autotrophic picoplankton (*Prochlorococcus* (Pro), low (LF-Syn) and high (HF-Syn) phycoerythrin fluorescence populations of *Synechococcus* and small (Speuk) and large (Lpeuk) picoeukaryotes). The first black-gray bar shows the same for heterotrophic prokaryotes (<100 m) while the second black-gray bar shows the values integrated through the entire water column (<700 m).

salinity values started to increase with depth (Fig. 8). There was no significant clustering of samples according to the different seasons (PERMANOVA: $r^2 = 0.04$, $p = 0.33$), indicating that the effect of depth layer was stronger than season (Fig. S4).

## DISCUSSION

The Red Sea represents a unique environment to investigate how picoplankton, the dominant planktonic size class at low latitudes (*Buck, Chavez & Campbell, 1996*; *Malmstrom et al., 2010*; *Olson et al., 1990*), respond to some of the highest natural temperatures and salinities that can be found in the ocean. We present here a comprehensive flow cytometric assessment of autotrophic and heterotrophic groups at both the vertical and seasonal scales at a 700-m deep station in the central Red Sea. Surface waters showed persistently high stratification, limiting the availability of DIN and DIP, which resulted in low phytoplankton biomass for most of the year (Figs. 1F and 2) and a clear dominance of small cells consistent with previous work (*Bock et al., 2018*; *Van den Engh et al., 2017*; *Wei et al., 2019*). Accordingly, DOC concentrations did not exceed 95 μmol L$^{-1}$. More information on the hydrological features and DOC dynamics of the study site can be found in *Calleja, Al-Otaibi & Morán (2019)*. In the nearby shallow waters of KAUST Harbor, although conditions were still oligotrophic year-round, higher concentrations of DIN and DOC were occasionally observed (*Silva et al., 2019*).

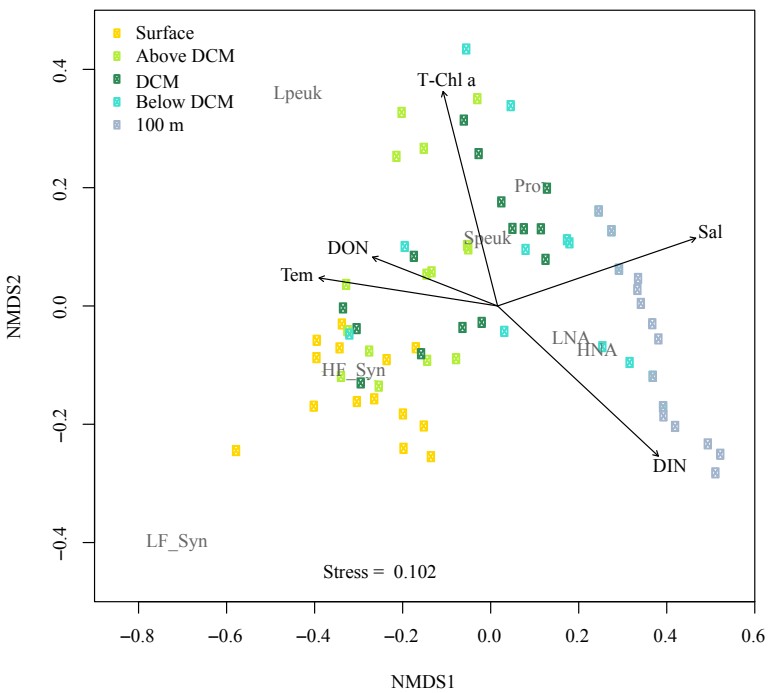

**Figure 8** **Nonmetric multidimensional scaling (NMDS) analysis of Bray–Curtis distances of the abundances of autotrophic and heterotrophic picoplankton and significant environmental variables in the upper epipelagic zone.** All samples were arranged by five depth categories (surface, above DCM, DCM, below DCM and 100 m) and coupled to significant ($p < 0.05$) environmental variables: temperature (Tem), salinity (Sal), total chlorophyll $a$ (T-Chl $a$), dissolved inorganic nitrogen (DIN) and dissolved organic nitrogen (DON) concentration. The centroids of the different populations (labeled in gray) indicate where each population was more abundant, and the vectors indicate the direction and strength of the environmental parameters. Group abbreviations as in Fig. 7 and in the main text.

## Vertical distribution of picoplankton

In this study, although cyanobacteria and picoeukaryotes made up most of the picophytoplankton biomass as chlorophyll $a$ (Fig. 2B), different depth preferences for each group were found. *Prochlorococcus* and the two size fractions of picoeukaryotes tended to show higher abundances at a depth of the DCM, which ranged from 40 to 76 m, than at the surface where both groups of *Synechococcus* peaked (Figs. 3A–3E). This distribution was further confirmed by the NMDS analysis of all samples (Fig. 8), showing a clear cluster of surface samples (with higher temperatures and DON) dominated by *Synechococcus*, while *Prochlorococcus* and picoeukaryotes dominated the DCM. This vertical segregation of *Prochlorococcus* and *Synechococcus* is well known (*Partensky, Hess & Vaulot, 1999*; *Rabouille, Edwards & Zehr, 2007*) and indicates a different adaptation to ambient light conditions. The light-harvesting antenna of *Synechococcus* have phycobilisomes with phycobiliproteins (phycoerythrin and phycocyanin) that confer them a higher ability to stand the high irradiances (including UV wavelengths) found at the surface (*Biller et al., 2015*). Several studies have suggested that *Prochlorococcus* is more sensitive to sunlight, particularly to UV potentially causing DNA-damage, than *Synechococcus* (*Agustí, 2004*;

*Boelen et al., 2000*; *Boelen et al., 2002*). Accordingly, *Prochlorococcus* is better adapted to capture the blue wavelengths that predominate deeper in the water column (*Biller et al., 2015*), thus giving rise to the observed differences in vertical distribution. However, the maximum depth at which we were able to detect cyanobacteria and picoeukaryotes by flow cytometry was generally 100 m. Molecular analysis is more sensitive than flow cytometry at finding rare populations. A recent study in the Red Sea assessing 16S rRNA gene sequences, more sensitive than flow cytometry, was able to find *Prochlorococcus* below 200 m, though with low numbers (*Shibl et al., 2016*). In other studies conducted in tropical waters, *Prochlorococcus* and *Synechococcus* were, however, detected by flow cytometry at depths of 150 to 200 m (*Bock et al., 2018*; *Partensky, Hess & Vaulot, 1999*; *Van den Engh et al., 2017*). The flow cytometer used in this study has a high sensitivity to detect small cells (<1 μm) and viruses (*Monier et al., 2017*). Thus, we believe that the apparent disappearance of picophytoplankton at those depths is rather a reflection of the already low numbers found in shallower depths compared with other studies than a problem with the detection limit (*Ribeiro et al., 2016*). Bearing in mind that 15 profiles along the annual cycle might still have missed the period of highest concentration, to our knowledge, the maximum abundance observed of *Prochlorococcus* ($1.63 \times 10^5$ cells ml$^{-1}$) lies among the lowest ever recorded. For instance, the maximum values in the subtropical and tropical oceans found during the Malaspina-2010 expedition were $14.2 \times 10^5$ cells ml$^{-1}$ for the Atlantic, $6.35 \times 10^5$ cells ml$^{-1}$ for the Indian and $3.27 \times 10^5$ cells ml$^{-1}$ for the Pacific (*Agustí et al., 2019*). The mean abundance of *Synechococcus* at the study site was also lower ($1.16 \times 10^4$ cells ml$^{-1}$) than in the Indian ($2.34 \times 10^4$ cells ml$^{-1}$) and Pacific ($4.85 \times 10^4$ cells ml$^{-1}$) oceans, but higher than in the Atlantic ($0.87 \times 10^4$ cells ml$^{-1}$). Very low abundances of picoeukaryotes were also consistently found in this study ($9.7 \times 10^3$ cells ml$^{-1}$) compared to the Atlantic ($18.5 \times 10^3$ cells ml$^{-1}$), Indian ($16.7 \times 10^3$ cells ml$^{-1}$) and Pacific ($79.3 \times 10^3$ cells ml$^{-1}$) oceans (*Agustí et al., 2019*). As in previous reports, consistent associations between the decrease in abundance and the increase in cell size and relative red fluorescence were observed for all picophytoplankton groups except for large picoeukaryotes (Figs. 3F–3O). This increase should be primarily attributed to the combined effects of depth-varying environmental variables such as inorganic nutrients availability and light (*Chen et al., 2011*), although shifts in species composition may also play a role (*Campbell & Vaulot, 1993*). The decrease in irradiance drives the need to synthesize more proteins and pigments to capture the fewer photons reaching the deeper layers (*Van den Engh et al., 2017*).

Regarding the vertical distribution of heterotrophic prokaryotes, we confirm the findings of two recent studies conducted at the same site as ours, focused on the interactions of bacteria with DOC stocks at the diel (*García et al., 2018*) and seasonal scales (*Calleja, Al-Otaibi & Morán, 2019*). As previously reported (*Calleja, Al-Otaibi & Morán, 2019*; *García et al., 2018*), LNA bacteria dominated in the epipelagic zone while HNA bacteria prevailed in the mesopelagic zone. The lower relative numbers of HNA cells in the upper 100 m (usually below 51%) could be explained by the presence of protistan grazers with a preference for the larger HNA cells (*Gonzalez, Sherr & Sherr, 1990*; *Lefort & Gasol, 2014*). Recent work has shown that the abundances of heterotrophic nanoflagellates were negatively correlated
with the sizes of both LNA and HNA cells, suggesting a preference to graze on the larger cells from both groups (Sabbagh et al., submitted). In turn, the dominance of HNA cells in the whole mesopelagic layer suggests either a release from grazing pressure (*Lara et al., 2017*) or that different taxa belonging to the HNA cluster are better suited to exploit the DOC compounds found at depth (*Calleja, Al-Otaibi & Morán, 2019*). The mesopelagic zone in this Red Sea site is characterized by a deep scattering layer, located between 400 and 600 m, where vertically migrating fish concentrate during the day (*Calleja et al., 2018*; *Røstad, Kaartvedt & Aksnes, 2016*). This layer seems to play an essential role in fast carbon transport and cycling by heterotrophic prokaryotes, as shown in previous studies (*Calleja, Al-Otaibi & Morán, 2019*; *García et al., 2018*).

Overall, the vertical distribution of picoplankton was most clearly affected by depth, in turn related to strong gradients in environmental variables (temperature, light, UV, inorganic nutrients, etc.), as clearly observed in the NMDS distribution of samples (Fig. 7) that cluster according to layer much more obviously than to season (Fig. S4). However, seasonal patterns became more evident when considering the depth-averaged or integrated values, as discussed below.

## Seasonal variation of picoplankton

Except at very high latitudes (*Cottrell & Kirchman, 2009*; *Li, 2009*; *Waleron et al., 2007*), cyanobacteria numerically dominate picophytoplankton communities, although the prevailing genus depends on the specific physicochemical properties and trophic structure. The dominance of *Prochlorococcus* has been frequently observed in high temperature, low nutrient and stratified waters, while *Synechococcus* and picoeukaryotes are usually predominant at lower temperatures, higher nutrient concentrations and more mixed waters (*Campbell et al., 1997*; *Malmstrom et al., 2010*). It has been hypothesized that the seasonality of picoplankton groups in tropical and subtropical oceans is less pronounced than in temperate or polar regions (*Bunse & Pinhassi, 2017*). However, although our site can be safely considered as permanently oligotrophic since it is strongly stratified year-round, surface temperature did indeed change between seasons (Table 2). On an annual scale, the longest subtropical time series at BATS displays high abundance of *Prochlorococcus* in summer and fall due to strong stratification and low values in late winter due to deep mixing events, while this pattern is much less visible at HOT (*Campbell et al., 1997*; *DuRand, Olson & Chisholm, 2001*; *Giovannoni & Vergin, 2012*; *Malmstrom et al., 2010*). A similar seasonal variability has also been reported in the Gulf of Aqaba in the northern Red Sea (*Al-Najjar et al., 2007*). However, two major differences were observed in this study. In spite of the overall dominance of *Prochlorococcus* especially noticeable in summer, *Synechococcus* unexpectedly outnumbered *Prochlorococcus* in winter and fall in the epipelagic layer (Fig. 5C). The fact that two populations of *Synechococcus* of differing orange fluorescence, LF-Syn and HF-Syn, were consistently found year-round did not result in major divergences in seasonality (Fig. 5B). Altogether, the total abundance of *Synechococcus* at our site peaked in winter the same as at HOT station (*Campbell et al., 1997*; *Malmstrom et al., 2010*), while the maximum abundance at BATS was found during the spring bloom when the mixed layer deepened and inorganic nutrients were detectable in

surface layers (*DuRand, Olson & Chisholm, 2001*). Picoeukaryotes have been reported to be more abundant in spring at both sites (*Campbell et al., 1997*; *DuRand, Olson & Chisholm, 2001*). In our dataset, although the two size fractions demonstrated different seasonality, picoeukaryotes generally tended to peak either at the beginning (winter-spring) or the end of the year (fall) (Fig. 5D).

With complete seasonal coverage, we confirm the finding that LNA heterotrophic prokaryotes dominate in the upper epipelagic (<100 m) while their HNA counterparts prevail in the mesopelagic zone (≥200 m) (*García et al., 2018*). Heterotrophic bacteria and archaea have been reported to present higher abundances in summer and decline in fall at BATS (*Carlson, Ducklow & Sleeter, 1996*) while the peak at HOT occurred in summer-fall (*Campbell et al., 1997*). In this study, the seasonality of both the LNA and HNA groups, as well as their sum, was less noticeable than autotrophic picoplankton, though low numbers were mostly observed in summer, as already reported by *Calleja, Al-Otaibi & Morán (2019)*. Bottom-up control by phosphorus could partially explain the decrease of heterotrophic prokaryotes in summer in the upper 100 m. *Calleja, Al-Otaibi & Morán (2019)* reported that epipelagic DIN:DIP ratios (without ammonium) peaked at 19 during summer, while lower values of ca. 11 were observed during the rest of the year, concomitant with DOC accumulation that could be a consequence of nutrient-limited and low standing stocks. Concurrently, in the experimental assessment of specific growth rates in the shallow waters of KAUST Harbor, top-down control by protistan grazers has been demonstrated to play an important role in regulating heterotrophic prokaryotes standing stocks (*Silva et al., 2019*).

There is little information about primary productivity (*Qurban, Wafar & Heinle, 2019*) and planktonic metabolism (*López-Sandoval et al., 2019*) in the Red Sea to allow an assessment of the seasonality of its metabolic balance (i.e., the periods of net autotrophy vs. net heterotrophy, *García-Martín et al., 2019a*; *García-Martín et al., 2019b*). We can still compare the respective biomasses of autotrophs and heterotrophs within the smaller size fraction, which collectively support to a large extent the higher trophic levels in oligotrophic environments. The relative importance of heterotrophic prokaryotes biomass to total planktonic biomass has been shown to increase with decreasing trophic state (*Azam, 1998*; *Biddanda, Ogdahl & Cotner, 2001*; *Del Giorgio, Cole & Cimbleris, 1997*; *Gasol, Del Giorgio & Duarte, 1997*), with an average ratio of 1.85 in the oligotrophic ocean (*Buck, Chavez & Campbell, 1996*; *Cho & Azam, 1990*; *Gasol, Del Giorgio & Duarte, 1997*). Considering only picoplankton, autotrophic cells make a higher contribution to total biomass in meso- to eutrophic areas, while heterotrophic bacteria and archaea typically become more important in tropical and subtropical oligotrophic oceans (*Harris, Duarte & Nixon, 2006*; *Regaudie-de Gioux & Duarte, 2013*; *Zhang, Jiao & Hong, 2008*). If we restrict our analysis to the first 100 m, autotrophic picoplankton biomass consistently exceeded that of heterotrophic prokaryotes biomass (Fig. 7), suggesting that the upper central Red Sea would be a net autotrophic ecosystem over the entire annual cycle (i.e., primary production would exceed community respiration), in agreement with the recent study of *López-Sandoval et al. (2019)*. The major contributor to autotrophic picoplankton biomass was *Prochlorococcus* as in other oligotrophic waters (*Wei et al., 2019*; *Zhang, Jiao & Hong, 2008*), except in

winter. However, if we extend the comparison between autotrophic and heterotrophic picoplankton biomass to the bottom of the study site, the ecosystem would then tend to net heterotrophic, but this difference was not very marked (Fig. 7). It is noteworthy that KAEC station lies between the metabolically balanced or net heterotrophic in the northern Red Sea and the net autotrophic waters of its southern reaches (*López-Sandoval et al., 2019*). In any case, further studies are necessary to fully understand the functioning of the central Red Sea pelagic ecosystem by a comprehensive assessment of its matter and energy fluxes.

This flow cytometry-based study is the first detailed temporal account of picoplankton abundance, single-cell characteristics and biomass covering from epi- to mesopelagic waters ever conducted in the central Red Sea. Future studies of the variations occurring at the daily scale will help interpret the seasonal patterns of autotrophic and heterotrophic picoplankton described here.

# CONCLUSION

This work presents different vertical segregation of the picoplanktonic groups surveyed. *Synechococcus* and LNA heterotrophic prokaryotes tended to occupy shallower layers than *Prochlorococcus*, picoeukaryotes and HNA heterotrophic prokaryotes. Seasonality was clearly depicted by the two genera of cyanobacteria, with *Synechococcus* exceeding *Prochlorococcus* cell numbers in early winter and late fall. Picoeukaryotes also tended to be more abundant in winter and fall, contributing to a seasonal structuring of picophytoplankton in Red Sea waters similar to higher latitude ecosystems. The seasonal patterns of heterotrophic prokaryotes were less noticeable than those of picophytoplankton and we did not find clear evidence of higher biomass of picoplanktonic heterotrophs at the study site year-round. Although the vertical gradients in environmental conditions had a major effect on the distribution of autotrophic and heterotrophic picoplankton, temporal changes over the year emerged as an important feature to be considered in future studies of the Red Sea pelagic ecosystem.

# ACKNOWLEDGEMENTS

We gratefully acknowledge the crew of the RVs Thuwal and KAUST Explorer and all the personnel from the Coastal and Marine Resources Core Lab for their diligent field-work assistance. Miguel Viegas aided enormously with fieldwork and sample collection, and he was also responsible for chlorophyll analysis.

## Funding

This research was supported by King Abdullah University for Science and Technology through the baseline funding (BAS/1/1069-01-01) provided to Xosé Anxelu G. Morán. The funders had no role in study design, data collection and analysis, decision to publish, or preparation of the manuscript.

## Grant Disclosures

The following grant information was disclosed by the authors:
King Abdullah University for Science and Technology through the baseline funding:
BAS/1/1069-01-01.

## Competing Interests

Xosé Anxelu G. Morán is an Academic Editor for PeerJ.

## Author Contributions

- Najwa Al-Otaibi conceived and designed the experiments, performed the experiments, analyzed the data, prepared figures and/or tables, authored or reviewed drafts of the paper, and approved the final draft.
- Tamara M. Huete-Stauffer analyzed the data, authored or reviewed drafts of the paper, performed non-metric multidimensional scaling (NMDS) of Bray-Curtis dissimilarity distances and pairwise PERMANOVAs, and approved the final draft.
- Maria Ll. Calleja analyzed the data, authored or reviewed drafts of the paper, provided and analyzed chemical data, and approved the final draft.
- Xabier Irigoien and Xosé Anxelu G. Morán conceived and designed the experiments, authored or reviewed drafts of the paper, and approved the final draft.

## Field Study Permissions

The following information was supplied relating to field study approvals (i.e., approving body and any reference numbers):

Sailing permission approved by Saudi Coast Guard.

## Data Availability

The raw data are available as a Supplemental File.

## Supplemental Information

Supplemental information for this article can be found online at http://dx.doi.org/10.7717/peerj.8612#supplemental-information.

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
