# Peer review of "Seasonal variability and vertical distribution of autotrophic and heterotrophic picoplankton in the Central Red Sea"

_PeerJ, doi:10.7717/peerj.8612_

## Round 0.1 · original submission · Minor Revisions

All three reviewers recommended your manuscript be published after minor revisions, please see their specific comments below (as well as the annotated PDF for Reviewer 2). In particular, they asked for the introduction to be significantly improved. I would also suggest the abstract be shortened to only include important findings of the manuscript.
Please provide a detailed point-by-point reply to each of the reviewers' comments.

I look forward to receiving the revised manuscript.

·

Basic reporting

Review report:
“Seasonal Variability and Vertical Distribution of Autotrophic and Heterotrophic Picoplankton in the Central Red Sea”

This study has investigated the vertical and seasonal variations of hydrological conditions and picoplankton communities in the central Red Sea, respectively.

Experimental design

Sampling plan in the present study is very good and the data set is also highly interesting. However, there are many repetitive descriptions and minor problems in the manuscript, I think the manuscript is needed to be compacted and revised. Overall, this manuscript is suggested to be published after aborative revisions. Specific comments are as follows:

Validity of the findings

The authors revealed that maximum cell abundance for each picoplankton group showed large differences in vertical distribution; while picoplankton abundances exhibited various patterns for seasonal variation. Consistent with previous observations, autotrophic picoplankton biomass in the upper waters accounted for the majority of total picoplankton biomass. The biomass of heterotrophic prokaryotes throughout the whole water column, however, were much higher than that of autotrophic picoplankton. Conclusively, the seasonal differences of picoplankton assemblages in the central Red Sea are not fundamentally different to higher latitude regions.

Additional comments

Comments to the Authors
Review report:
“Seasonal Variability and Vertical Distribution of Autotrophic and Heterotrophic Picoplankton in the Central Red Sea”
General comment:
(1)This study has investigated the vertical and seasonal variations of hydrological conditions and picoplankton communities in the central Red Sea, respectively. The authors revealed that maximum cell abundance for each picoplankton group showed large differences in vertical distribution; while picoplankton abundances exhibited various patterns for seasonal variation. Consistent with previous observations, autotrophic picoplankton biomass in the upper waters accounted for the majority of total picoplankton biomass. The biomass of heterotrophic prokaryotes throughout the whole water column, however, were much higher than that of autotrophic picoplankton. Conclusively, the seasonal differences of picoplankton assemblages in the central Red Sea are not fundamentally different to higher latitude regions.
(2) Sampling plan in the present study is very good and the data set is also highly interesting. However, there are many repetitive descriptions and minor problems in the manuscript, I think the manuscript is needed to be compacted and revised. Overall, this manuscript is suggested to be published after aborative revisions. Specific comments are as follows:
ABSTRACT:
(1)P22-23, please change “standing stocks and cellular characteristics” into “abundances and biomass”;
(2)P23, please delete the repetitive word “cyanobacteria”, or change “Prochlorococcus and Synechococcus” into “picocyanobacteria”;
(3)P27-29, I don't understand why the authors described the hydrological conditions starting at “The sea surface temperature ranged from.......” as one part of the abstract, and what the significance and correlation are between these sentence and the manuscript's aim in this part.
(4)Additionally, P29-45, these sentences are mainly repetitive descriptions of the results and some are redundant, please simplify.
(5)P50-52, please check English grammar
Key words:
(6)The authors should add the key words, which are important for readers to know more about this paper.
INTRODUCTION:
(7)P62, please change “cyanobacteria” into “picocyanobacteria”
(8)P109, please change “cellular characteristics” into “biomass”
MATERIALS AND METHODS:
(9)P129, 200 ml water samples? Is it enough for acquiring the size-fractionated Chl a?
(10)P141, 10% paraformaldehyde? Please check;
(11)P150, Bibliographic references should be added to support the application of SYBR Green II;
(12)P151, A recently published paper showed the detection of surface Prochlorococcus is a major challenge to even the most sensitive instruments (BD Influx with small particle detector; Ribeiro C G et al., 2016, Estimating microbial populations by flow cytometry: Comparison between instruments), and I’m not sure your instrument has a small particle detector, if not, please give full details in the Instrument Settings. Please refer to Wei et al., 2019, Dynamic responses of picophytoplankton to physicochemical variation in the eastern Indian Ocean.
(13)P159, I am confused that how the authors calculate this conversion factor 237 fg C μm–3, Please list the calculation formulas and references.
RESULTS:
(14)P177, Figures quality need to be improved, especially for the font format (all should be changed into Times New Roman). Please provide high resolution graphs;
(15)P178, Please delete the redundant word “sea”;
(16)P200, Please use seasonal interval, rather than days of year;
(17)P279, I don't understand the meaning of this sentence starting at “The stress level was relatively low, indicating a reliable distribution of samples in two arbitrary dimensions.” How do you assess the reliability based on stress level? Please explain and list bibliographic references.
(18)P281, Please list the original data of the correlation coefficient between NMDS scores and environmental variables, and please explain “r, r2 and p”
(19)P285-288, I don't understand what the authors would like to express through these sentences, is it correlation? If so, where is the correlation coefficient?
DISCUSSION:
(20)P293, Please list subheadings in the discussion section;
(21)P370, DIP is a key factors in affecting the distributions of picoplankton, but i am confused why the authors did not analyze the relationship between DIP and picoplankton abundance based on the NMDS analysis in the discussion section;
(22)P371, The authors indicated that the vertical distribution of picoplankton was most clearly affected by depth, and is related to gradients in environmental variables, for example light irradiation; however, I don't understand why the authors presented the relationship between picoplankton and light as per bibliographic references, instead of analyzing the field irradiance (PAR) acquired from SeaBird SB9 Plus or IDRONAUT 305 CTDs.

Reviewer 2 ·

Basic reporting

The manuscript was very well written and the story, logic and results easy to follow.
However, I felt the first two paragraphs of the introduction were rather long and it took a long time to get to the third paragraph where reasoning for the study was first introduced. Suggest to make the first two paragraphs more concise, and highlight the gaps that are relevant to this study (i.e oligotrophic environment, high temperature and salinity compared to other regions, limited information of picoplankton in these types of environments). Then using those gaps, lead into the third paragraph to highlight what is known of picoplankton in the Red sea and why having an understanding of temporal changes in their abundance is important. I felt the link to upper food webs in the Red Sea was lacking – suggest to include more information on this (in a few sentences).
The structure of the manuscript conforms to PeerJ standards.
Raw data is supplied in the supplementary information.
Figures are well presented, labelled well and on the most part easy to understand. A few suggestions provided below to improve figures and their interpretation by readers.
Figure 3– suggest to have Synechococcus (LF and HF) on the same scale, and Speuks and Lpeuks on the same scale – to allow clear interpretation by the reader.
Figure 4 – Have D and E on the same scale.
Figure 6 – (cell ml-1) change to (cells ml-1)

Experimental design

The research was well defined. The way the research fills an identified knowledge gap could be made clearer, in the abstract, introduction and discussion. See above for comments on the intro in relation to this and expand on this briefly in abstract and further in discussion.
The periodic seasonal sampling provided for meaningful and provided for clear results. Measured parameters were valid for the study and explained well in the methods section.
The statistical analyses performed for the study were appropriate for the data, allowing for detection of differences in the picoplankton community between seasons and in the vertical water column where found. These results were well detailed and depicted.

Validity of the findings

The findings of the study are novel for the region of the Red Sea and expand on the increasing information available on picoplankton in the ocean. The trends were displayed well in figures and interpreted well in the manuscript. Discussion points and speculation and conclusions reached were for the most part, valid. Some discussion points were highlighted in the pdf version to either rework, condense or add more information. See pdf for this.

Additional comments

This study provides temporal and in situ vertical distribution information on picoplankton in the oligotrophic, high temperature and salinity waters of the Red Sea. Distributions were linked to environmental parameters, and showing light and nutrients to be clear drivers of the picoplankton community. This study fills a knowledge gap for the region of the Red Sea and what is known on the temporal and vertical picoplankton abundances and distributions.
In addition to the below review, please see also attached manuscript with edits/and suggestions to improve the manuscript. Most suggested changes and additions are included in the pdf as comments and word edits.
In all this manuscript provides new data in relation to picoplankton in the Red Sea. The survey design well thought out and instigated, and all the data collected was relevant to the writing of the manuscript. The manuscript was well written, clear and concise in most parts (the suggested changes/edits are indicated in the pdf). In all, with the suggested edits, I would recommend publication of this manuscript to PeerJ.

Annotated reviews are not available for download in order to protect the identity of reviewers who chose to remain anonymous.

Reviewer 3 ·

Basic reporting

The MS by Al-Otaibi et al (#39703) addresses the question of the seasonal variability and vertical distribution of autotrophic and heterotrophic picoplankton in the central Red Sea. It provides the first insight into temporal dynamics of ecologically relevant players.

The article is well written and the work presents an interesting dimension to a less studied marine ecosystem.
However, the introduction reads like a review of existing literature, including their own, rather than a structured argument for the study. This part needs significant improvement; else, it digresses the readership.

Some terminologies are not clear (e.g., “unicell”).
Important literature from recent work is also not referenced though it represents significant advancement to the functional molecular ecology of the Red Sea (Thompson et al. 2016. The ISME Journal 11, 138-151
; 2013. Ecology and Evolution 3). For instance, in line #95

In line#108, I would suggest re-phrase the permanently stratified statement, by qualifying with a specific depth. Stratification is defined by depth of the water column, and this changes considerably by season. Indeed, line#304–305 shows that the nutrient dynamics change in this site. So up to what depth is the Red Sea water (at the site) stratified?

Experimental design

Although the methods are well defined, with majority coming from the authors’ lab(s), it would help if a supplementary graphical figure were provided encompassing a guide to these approaches. At the moment it’s too verbose; and a figure would help navigate the experimental design better.

Validity of the findings

All data are provided and statistically sound.

In the abstract and line #298 please rephrase the word “700 m deep” to “700-m deep”; the former suggests that the sampling was done at 700 m!

In line#324, it should be pointed out that the two methods have different sensitivities; PCR is more sensitive; so that the oddity suggested is invalid. The subsequent line#330 only suggests differences in flow cytometer detection limits.

The line#331, the authors lay emphasis on the “lowest” abundance ever recorded. I think this is rather speculative since biomass in the ocean is dynamic, and can be low or higher depending on the season. So the values reported are only a steady-state situation at the time of sampling. To make the “lowest ever recorded” claim requires high-frequency sampling (s.f. line#437–438).

Additional comments

The MS is sound and the data provided is relevant for understanding ecosystem function of the changing pelagic global ocean. And studies coming from the Red Sea provide a basis for understanding the implications.
In sum, I recommend the acceptance of the MS, provided the minor issues are resolved.

---

## Round 0.2 · Minor Revisions

Thank you for the revised version of your manuscript, and for your response to reviewers that very satisfactorily answered their comments, as also confirmed by Rev. 1. I am sending this back to you for minor revisions again, simply because the paper would benefit from more of your answers to the reviewers (mostly Rev 1) being incorporated into the manuscript. Readers will likely have similar questions as the reviewers did, and so reviewers’ questions should be addressed in the paper, not simply in your response to reviewers. These edits are very minor as you already addressed them in your response to reviewers, and should be very easy to incorporate. Please also unabbreviate keywords (see comment by Rev 1). Please update the manuscript accordingly and I will be happy to accept the paper.

More specifically:
- you wrote (to Rev 1) “The flow cytometer we used (FACSCanto II, BD-Biosciences) has a very good sensitivity to detect particle sizes from 0.5 to 50 µm and even smaller than that since it has been routinely used for counting viruses.”. This information should be included in the paper, along with a reference to studies using it to count viruses.
- you wrote (to Rev 1) an explanation of NMDS and what stress represents. Please keep in mind that readers will not all be familiar with NMDS and stress; I would suggest including a sentence or two providing additional information (similar to your Rev 1 response) in section “Relationships with environmental variables” along with some reference(s).
- you wrote (to Rev 1) “the centroids of the different populations (labelled in gray) indicate where each population is more abundant and the vectors indicate the direction and strength of the environmental parameters”. This information would be very relevant in Fig. 8 caption (in particular, there is at present no mention in text of what the grey labels represent).
- you wrote (to Rev 1) “Initially, the NMDS analysis, included all environmental variables that were available (temperature, salinity, DIN, DIP, DOC, Chl a, stratification index (SI), depth of the upper mixed layer (UML), euphotic zone (Zph), DCM, and nutricline). However, some of these variables did not show a significant effect on the distribution of the samples, and were removed from the final figure.”. This is useful information that should appear in text, maybe simply mentioning in Fig. 8 caption that other variables (such as DIP, etc) were considered but found to be not significantly related to the distribution of samples and thus were not plotted.
- you wrote (to Rev 1) “we do not have the PAR data for all sampling times (only 7 out of 15)”. This information should appear in the “Material & Methods” section that currently reads as if all environmental properties are available for all casts.

A few additional minor comments:
- l. 57 should be “Picocyanobacteria” not picoyanobacteria
- l. 75 either “less is known” or “they are less known”
- in numerous places, parentheses are opened but not closed, see l. 81, 83, 101, 324, 346, 431
- in a couple of places, references are written as (XX YY) instead of XX (YY), see l. 311, 421
- l. 288: the correlation does not “result in” significant effects of temperature etc, but rather “indicates”
- l. 319 should be “the depth of the DCM”
- l. 398 HOT not HOTS

·

Basic reporting

no

Experimental design

no

Validity of the findings

no

Additional comments

Keywords should not be abbreviated! As the authors have considered all comments and suggestions, I recommend this MS for publication. Congratulations!

---

## Round 0.3 · accepted · Accept

Thank you for incorporating the suggested minor revisions. I am happy to approve this manuscript for publication.